# Particle-in-Cell Simulations for the Improvement of the Target Erosion Uniformity by the Permanent Magnet Configuration of DC Magnetron Sputtering Systems

Young Hyun Jo [1,2], Cheongbin Cheon [1], Heesung Park [1] and Hae June Lee [1,*]

1   Department of Electrical Engineering, Pusan National University, Busan 46241, Republic of Korea
2   Mechatronics Research, Samsung Electronics Company, Ltd., Hwaseong 18448, Republic of Korea
*   Correspondence: haejune@pusan.ac.kr

**Abstract:** Improving the target erosion uniformity in a commercial direct current (DC) magnetron sputtering system is a crucial issue in terms of process management as well as enhancing the properties of the deposited film. Especially, nonuniform target erosion was reported when the magnetic flux density gradient existed. A two-dimensional (2D) and a three-dimensional (3D) parallelized particle-in-cell (PIC) simulation were performed to investigate relationships between magnetic fields and the target erosion profile. The 2D PIC simulation presents the correlation between the heating mechanism and the spatial density profiles under various magnet conditions. In addition, the 3D PIC simulation shows the different plasma characteristics depending on the azimuthal asymmetry of the magnets and the mechanism of the mutual competition of the E × B drift and the grad-B drift for the change in the electron density uniformity.

**Keywords:** DC magnetron sputtering; particle-in-cell simulation





## 1. Introduction

Direct current magnetron sputtering (DCMS) is a standard physical vapor deposition (PVD) technology for metal deposition in a semiconductor process. The DCMS system applies DC electric power to a negatively biased metal target, whose atoms are sputtered by the ion bombardment. The atoms move through the chamber before they are deposited on a substrate on the opposite side. DCMS systems are usually operated at pressures lower than a few mTorr to minimize disturbance to diffusion and deposition of sputtered atoms. When the gas pressure is high, the sputtered atoms collide with neutrals, making the deposition inefficient. On the contrary, it is difficult to discharge in a low-pressure condition because the breakdown voltage rapidly increases on the left-hand side of Paschen's breakdown curve for a low-pressure regime. In order to keep a stable discharge under low-pressure conditions, a static magnetic field is applied to reduce electron transport in the perpendicular direction. In the parallel direction to the magnetic field, the magnetic mirror effect enhances the electron confinement, and finally, the plasma density is high at a particular location above the target. However, the magnetic field should not be too high to reduce the ion bombardment on the target. That is to say, the magnetic flux density should be in the range to magnetize not only electrons but also ions. Therefore, the design of the magnetic field profile under a given gas pressure and device structure is a critical issue. Many studies have been dedicated to revealing the physics of DCMS plasmas [1–6].

A standard DCMS system contains a cathode with a metal sputtering target and permanent magnets. The magnetic fields of permanent magnets make charged particles rotate and confine them to the circular motion commonly known as gyromotion. Magnetic confinement is the reason why DCMS systems sustain high-density plasmas at such low pressures. Plasmas consist of magnetically confined electrons, called magnetized electrons, and unmagnetized ions. They are generated near the sputtering target in DCMS systems.

Then, energetic ions accelerated in the sheath region of the plasmas, bombarding and sputtering atoms of the target materials. The sputtered atoms are deposited on a substrate. They can also be ionized while passing through plasma. The source of the deposited particles in a DCMS system is the ion bombardment on the target, which is controlled by the density profiles of the magnetized electrons. Therefore, it is crucial to understand the mechanism of plasma formation in the DCMS system.

We primarily need to design or modify components related to DCMS plasmas, such as the system length between a target and a substrate, target radius, magnetic configurations, and so on. The magnetic configuration is one of the most critical parameters to control the characteristics of the process results, including the target erosion uniformity. The conventional balanced magnetrons confine plasmas near the target well. In contrast, the unbalanced magnetrons can cause plasmas to escape toward the substrate since the magnetic field lines from the stronger magnets are not fully closed with the weaker magnets [1,4,5]. However, the specific characteristics and spatial distributions of plasmas depending on the magnet design are not so easy to anticipate, even though they are necessary for costly and closed processes such as semiconductor processing.

The magnetic field intensity of a DCMS system is typically a few hundred Gauss. The radius of the gyromotion, which is called the gyro-radius or Larmor radius, is $mv_\perp/|q|B$, where $m$ is the mass of the charged particle, $v_\perp$ is the velocity perpendicular to the direction of the magnetic field, $q$ is the electric charge of the charged particle, $B$ is the intensity of the magnetic field. In a typical DCMS system, the gyro-radius of an ion is tens of centimeters, while the gyro-radius of an electron is an order of a micrometer. It means that most of the ions in DCMS plasmas are not magnetized since the system length of a standard DCMS system is between a few centimeters and tens of centimeters, while electrons are magnetized and confined in the system. Therefore, the magnetized electrons directly affect the ionization region and the spatial distribution of DCMS plasmas. DCMS plasmas were simulated with fluid approaches [7–10], kinetic approaches [11–24], or hybrid methods [6,7,25–29]. However, the validity of fluid models is limited by gas pressure, which should be high enough to apply the fluid assumptions. That is why a kinetic approach is necessary to simulate low-pressure DCMS plasmas. Some previous studies used only the trajectories of electrons to reduce computational load without coupling the charged particle dynamics with the self-consistent field solver [30–34].

Most of the previous research has chosen the particle-in-cell (PIC) Monte Carlo collisions (MCC) method to simulate low-pressure DCMS plasmas. A PIC-MCC is a computational methodology to simulate plasma discharges [35]. The interactions between charged particles and the electromagnetic field are simulated by repeatedly calculating the Newton-Lorentz equation for the charged particle motion and the Maxwell equation for the electromagnetic field. The Poisson equation can be used instead of the Maxwell equation to obtain the electric field in the electrostatic system. MCC is an efficient method to calculate collisions with a preprocess of computing particles to collide in the PIC simulation, proposed by Vahedi et al. [36]. A full PIC-MCC simulation includes no assumptions about energy distributions in plasma dynamics and is thus considered the most accurate way of running a plasma simulation to date. Highly accelerated PIC-MCC simulations by graphics processing unit (GPU)-based parallelization have recently been reported [37–39].

As DCMS is a well-established technique, most fundamental physics are well understood, except for the effect of 3D magnetic field structure, which requires a three-dimensional (3D) PIC simulation. However, a 3D PIC simulation is still challenging and time-consuming because of the realistic size of the DCMS device. In this study, we investigate the effect of the curvature variation in 3D magnetic fields. Using the PIC-MCC method, this paper focuses on the fundamental relationship between the physics of DCMS plasmas and the magnetic field configurations of permanent magnets in the DCMS system. First, the influences of several actual magnetic field configurations, including asymmetric magnets for DCMS plasmas, are investigated with a two-dimensional (2D) PIC-MCC simulation as a preliminary study in Section 2. Then, the effects of azimuthal asymmetry and asymmetry

of permanent magnets on DCMS plasmas are discussed in more detail with a 3D PIC-MCC simulation in Section 3. Finally, the conclusion is followed in Section 4.

## 2. Two-Dimensional Simulation for the Variation of Magnet Configurations

This section elaborates on the effects of magnetic flux density with various configurations of permanent magnets using a 2D electrostatic PIC-MCC simulation (version 2023.02.01). The computational details are presented in Section 2.1. The effects of magnetic field magnitude on DCMS plasmas are discussed as a fundamental study in Section 2.2. The influence of the thickness of the magnetic yoke is discussed in Section 2.3. Finally, the effects of asymmetric magnets are investigated in Section 2.4.

### 2.1. Computational Details

An in-house 2D electrostatic PIC-MCC simulation code is utilized here [37]. The code is parallelized using CUDA and performed with an NVIDIA GeForce RTX 3090 (Gigabyte Technology Co., Ltd., New Taipei, Taiwan). This code calculates the charged particle motions for loops based on the cell number. It means that the simulation needs the dynamic load balance to improve the calculation speed for spatially nonuniform DCMS plasmas. However, the computation speed is fast enough to obtain each 2D result shown in this study within 24 h. A schematic diagram of the simulation domain is depicted in Figure 1. It represents a typical DCMS system with a copper sputtering target, which is also a cathode in this system. The applied electric power of the cathode is set to 100 W. The other conductors at the boundaries are all grounded and disconnected from the cathode, with a tiny gap between them. The upper cathode in the domain is covered with the substrate. The gas pressure is 1 mTorr of argon feed gas, which is considered uniform in the background of the discharge region for simplicity. Only argon plasmas are simulated without consideration of both sputtering processes and metal plasmas generated by sputtered atoms to focus on the influence of magnetic field configurations on the plasma profile. The ion-induced secondary electron emission coefficient of the copper target is assumed to be 0.2. The number of cells is $512 \times 180$, and the time step $\Delta t$ is $10^{-11}$ s in the simulation. The applied magnetic field is calculated using a freeware program called Finite Element Method Magnetics (FEMM) (version 4.2, 2019).

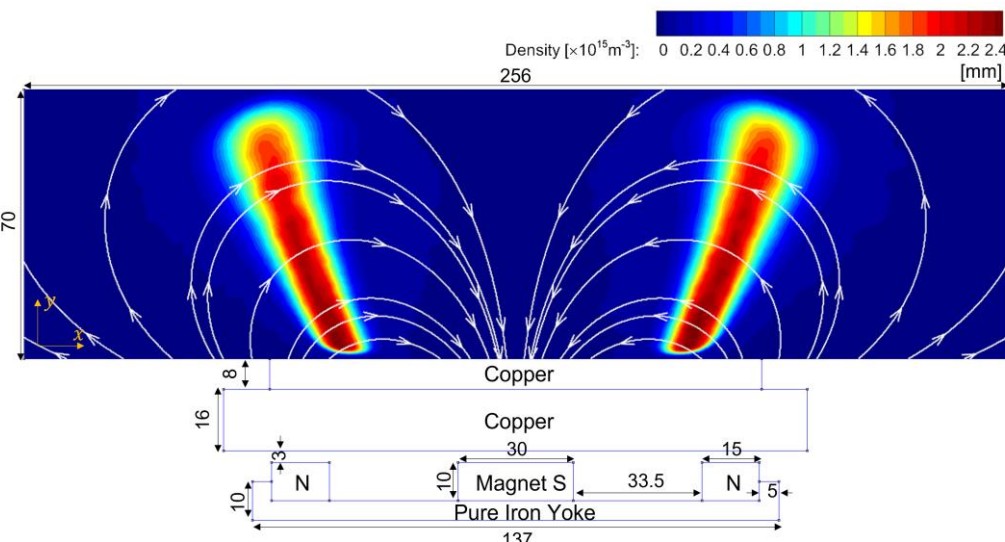

**Figure 1.** A schematic domain of the two-dimensional simulation with the plasma density and magnetic field lines. The length scale is in mm.

### 2.2. Influences of Magnetic Field Intensity

At first, two cases were selected to investigate the role of the magnitude of the magnetic flux density. Here, the configuration of the permanent magnets is the same, but only the

magnitude is different. The profiles of magnetic fields are depicted in Figure 2, where the maximum intensity of magnetic fields is 350 G in Figure 2a and 500 G in Figure 2b. They have the same magnetic field profiles, but the color bar scales and maxima differ for both figures. The height of the yoke is 10 mm, and the widths of the magnets are 30 mm for the south pole and 15 mm for the north pole. The gap distance from the north pole to the south pole is 33.5 mm and symmetric.

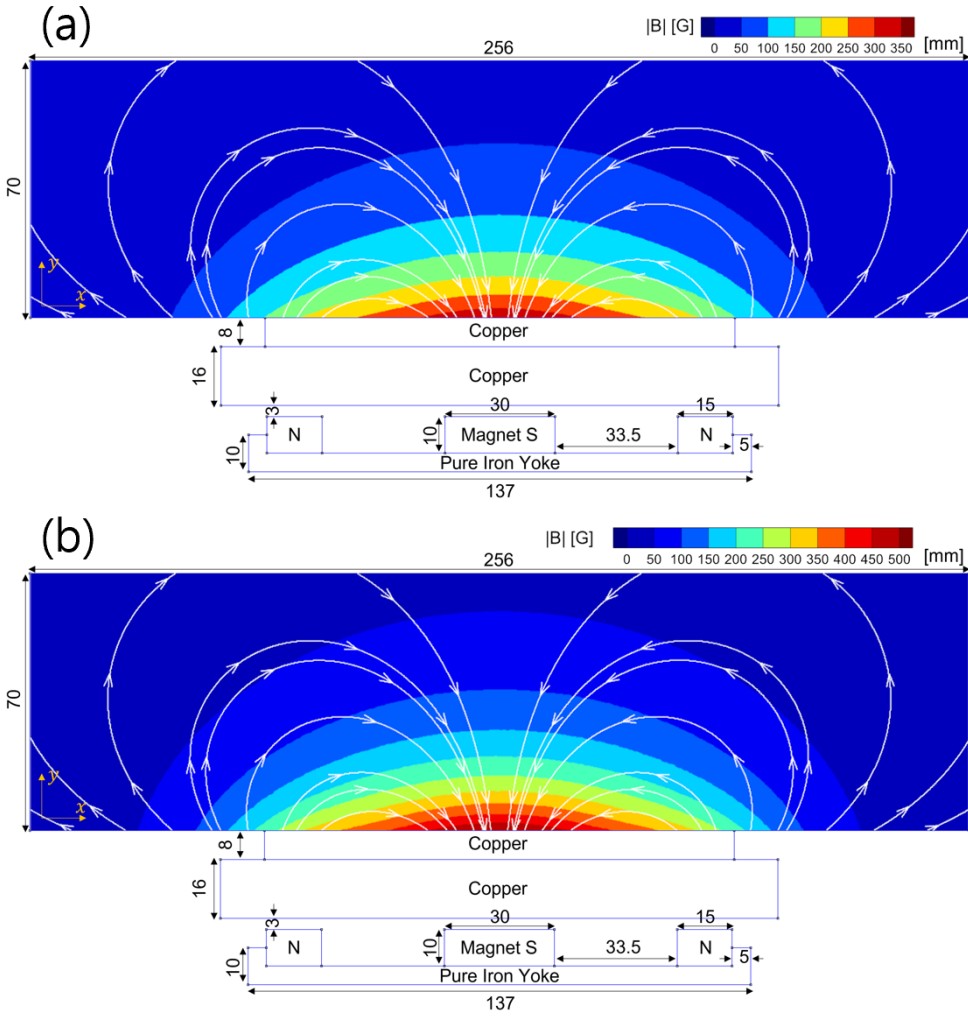

**Figure 2.** Magnetic fields with the maximum magnetic flux density of (**a**) 350 G and (**b**) 500 G.

The results of the PIC-MCC simulation are shown in Figure 3 for the plasma density and potential profiles. Only half of the domain is depicted here because the profiles are symmetric. The plasma density of the 350 G case is higher than that of the 500 G case. It is an unexpected result, as it is usually thought that stronger magnetic fields would confine plasmas better. Comparing Figure 3c,d, the sheath thickness in the 500 G case is broader than in the 350 G case since the plasma density is lower with the same target voltage of −470 V.

Figure 4 shows the Ohmic electron heating, $J{\cdot}E$, and the electron temperature $T_e$, where $J$ is the electron current density, and $E$ is the electric field intensity. Figure 4a shows enhanced electron heating in a broader space for the 350 G case compared with Figure 4b for the 500 G case. The electron temperature is higher near the substrate than the target, as shown in Figure 4c,d. Most ionizations occur in the high electron temperature region near the substrate. The source of the energetic electrons is the ion-induced secondary electron emission (SEE) from the target. They are accelerated by the electric field inside the sheath in front of the target to ionize the neutral gas or pass through the magnetized region until they

meet the sheath in front of the substrate. These electrons make the high-temperature region near the substrate. The ions generated there are accelerated directly toward the target to induce SEE without collisions at very low pressure. That is why the plasma density of the 500 G case is lower even though it has stronger magnetic fields and higher electron temperatures near the target than that of the 350 G case. It indicates that more ionization near the substrate is the key to enhancing the DCMS discharge at very low pressure. Another noticeable difference is that heated electrons in the 350 G case are the primary reason for the striation. The fundamentals of this phenomenon are not fully analyzed here, but an additional electron heating mechanism exists in addition to the γ-mode in typical DC or DCMS plasmas, unlike the 500 G case. The striation of DCMS plasmas is barely found at pressures higher than a few mTorr, as shown in the previous study with a similar magnetic condition [20]. The striation phenomenon may have a sudden transition regime depending on the magnetic field intensity, which will be investigated more in future work.

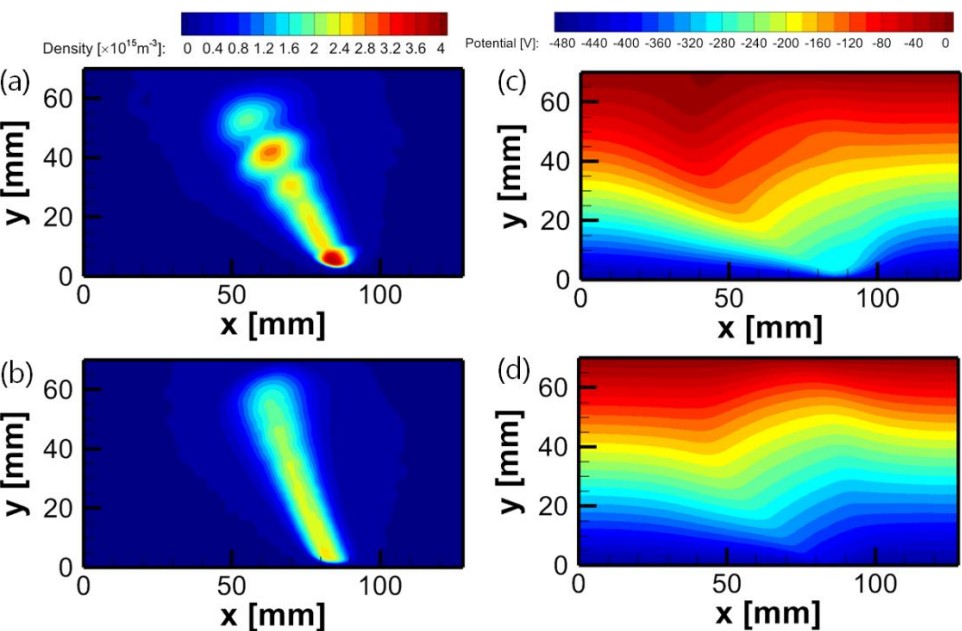

**Figure 3.** (**Left**) Plasma densities with the maximum magnetic flux densities of (**a**) 350 G and (**b**) 500 G. (**Right**) Electric potentials with the maximum magnetic flux densities of (**c**) 350 G and (**d**) 500 G.

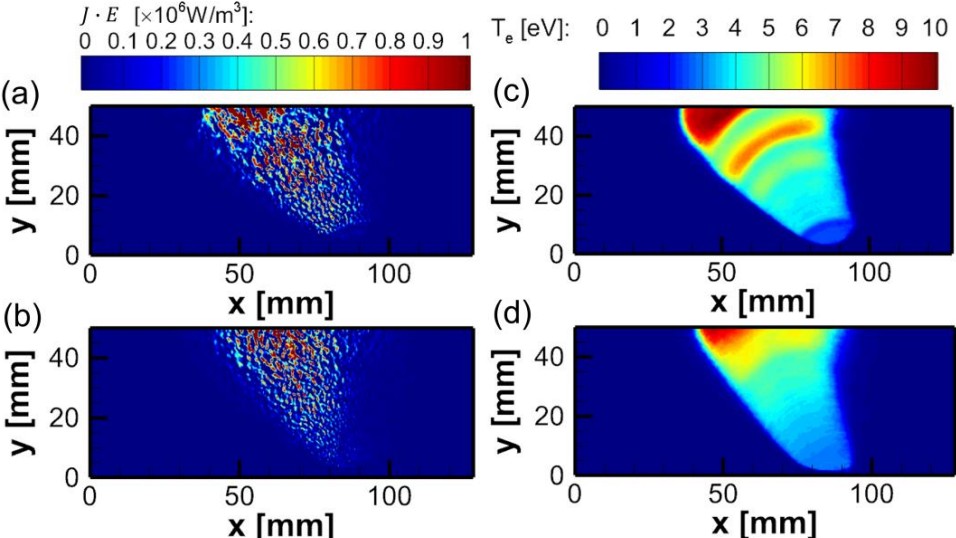

**Figure 4.** Profiles of $J \cdot E$ with the maximum magnetic flux densities of (**a**) 350 G and (**b**) 500 G. Profiles of electron temperature with the maximum magnetic flux densities of (**c**) 350 G and (**d**) 500 G.

### 2.3. Influences of the Yoke Thickness

Magnetic yokes commonly consist of ferromagnetic materials with high permeability and focus magnetic fluxes in the desired direction or position. In a DCMS system, magnetic yokes make magnetic fluxes from permanent magnets concentrate in the direction toward plasmas. The concentrated magnetic fluxes enhance the magnetic fields and the density of plasmas. The effects of various yoke configurations, including a wide yoke that moves the region of target erosion, were previously reported [38]. It shows that it is possible to control plasmas and target erosion profiles using the yoke magnets. In this section, the effects of the yoke thickness are discussed. The magnetic field lines and the magnitude of the magnetic flux density, with a thickness of the magnetic yoke of 15 cm, are depicted in Figure 5. The thicker yoke pulls the magnetic field slightly more toward the target, enhancing the magnetic field near the target.

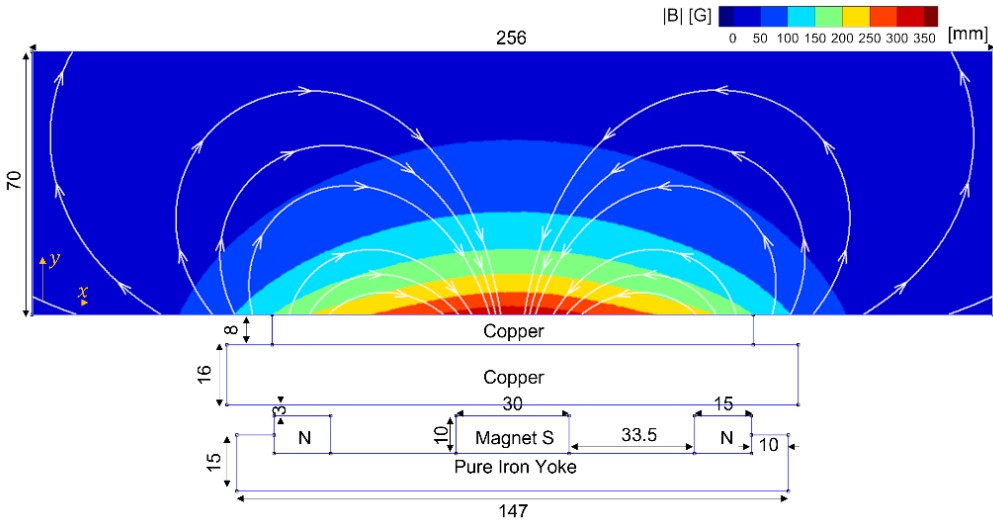

**Figure 5.** Magnetic field lines and the magnitude of the magnetic flux density of 350 G case with a magnetic yoke with a thickness of 15 mm.

The effects of the yoke thickness on DCMS plasmas are presented in Figure 6 for the plasma density and potential profiles and in Figure 7 for electron heating and the electron temperature. Compared with the change in the magnitude of the magnetic flux density, the plasma density near the target seems unaffected by the yoke thickness. However, the density near the top substrate is higher with the thicker yoke, as shown in Figure 6b. On the other hand, the electron temperature increases with the overall thickness of the yoke. The enhanced discharge with the thicker yoke can be explained by the increasing electron temperature observed in the middle region of the domain, as shown in Figure 7d. The striation patterns are related to the spatial variations of electron temperature caused by electron heating, as shown in Figure 7b. The stronger magnetic fields created by the thicker yoke trap more electrons, even far from the target. As a result, more ionization occurs with enhanced electron heating. Therefore, the yoke thickness can be one of the parameters to control the electron temperature and the density in the region a little bit away from the target. However, the strong magnetic field still limits the perpendicular transport of electrons near the target, and thus the density does not change significantly near the target for $0 < y < 10$ mm.

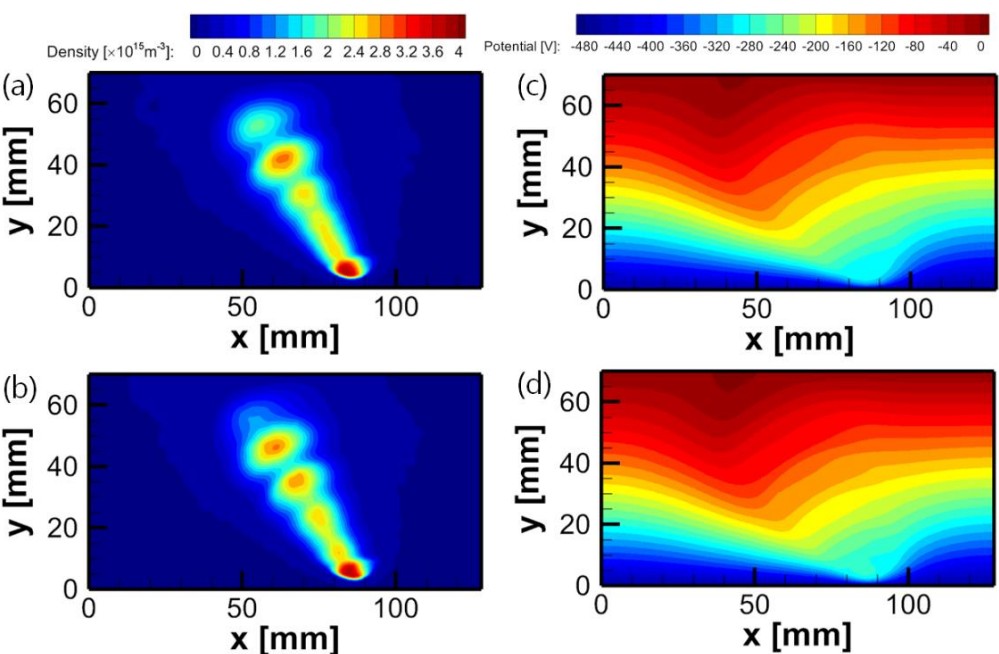

**Figure 6.** Plasma densities with the yoke thickness of (**a**) 10 mm and (**b**) 15 mm. Electric potentials with the yoke thickness of (**c**) 10 mm and (**d**) 15 mm.

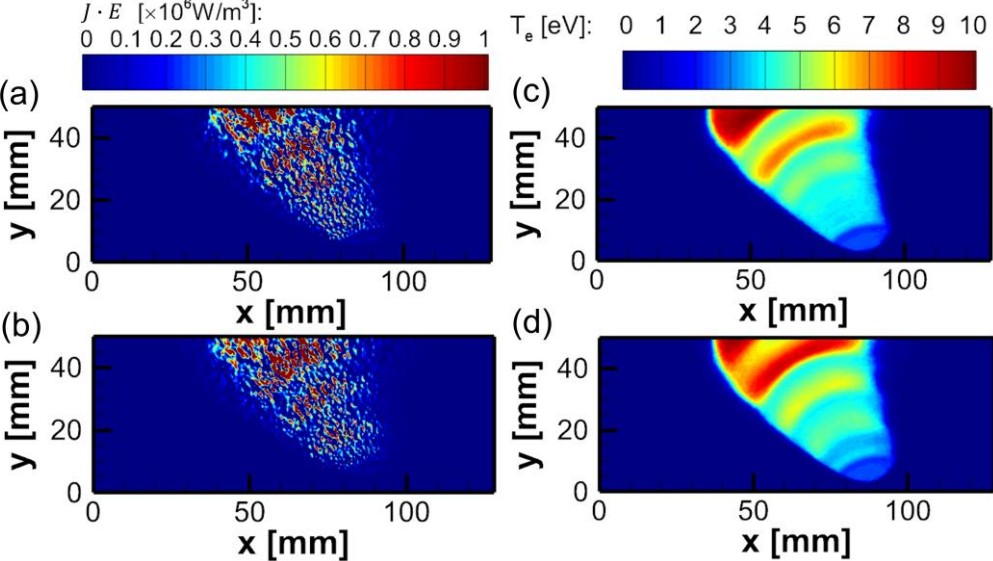

**Figure 7.** Profiles of $J \cdot E$ with the yoke thickness of (**a**) 10 mm and (**b**) 15 mm. Profiles of electron temperatures with the yoke thickness of (**c**) 10 mm and (**d**) 15 mm.

### 2.4. Effects of Asymmetric Magnets

A case of asymmetric permanent magnets, where the magnet on the right side is more intensified, is investigated in this section. The magnetic fields in this case are shown in Figure 8. In this case, the magnetic field on the right side with a wide magnet and short gap is stronger than the one on the left side with a narrow magnet and a long gap. The maximum magnitude of the magnetic flux density is set to be 350 G in this case.

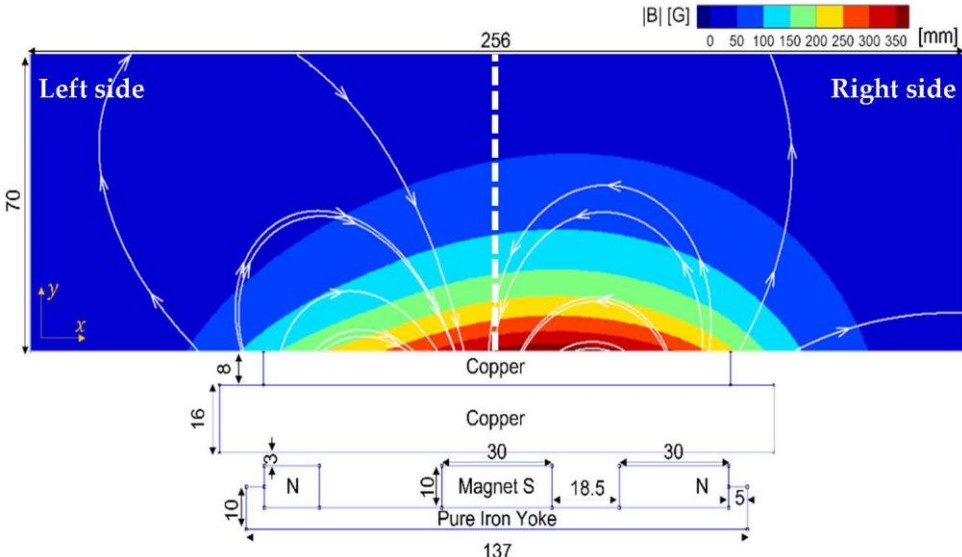

**Figure 8.** Magnetic field lines and the magnitude of the magnetic flux density for asymmetric magnets where the widths of the magnets are 15 mm on the left side and 30 mm on the right side.

The effects of asymmetric magnets on DCMS plasmas are presented in Figure 9 for the plasma density and electric potential profiles and in Figure 10 for electron heating and the electron temperature. The stronger magnetic fields on the right side confine plasmas in a similar shape to the 500 G case, while the weaker magnetic fields on the left side confine plasmas in a similar shape to the 350 G case. The lower density in the region where the magnetic fields are stronger can be explained in the same way as discussed in Section 2.2, although the striation is slightly less visible on the left side. Even though the electron confinement is better on the right side, the plasma density is high on the left side because the electron heating is enhanced there, as shown in Figure 10a.

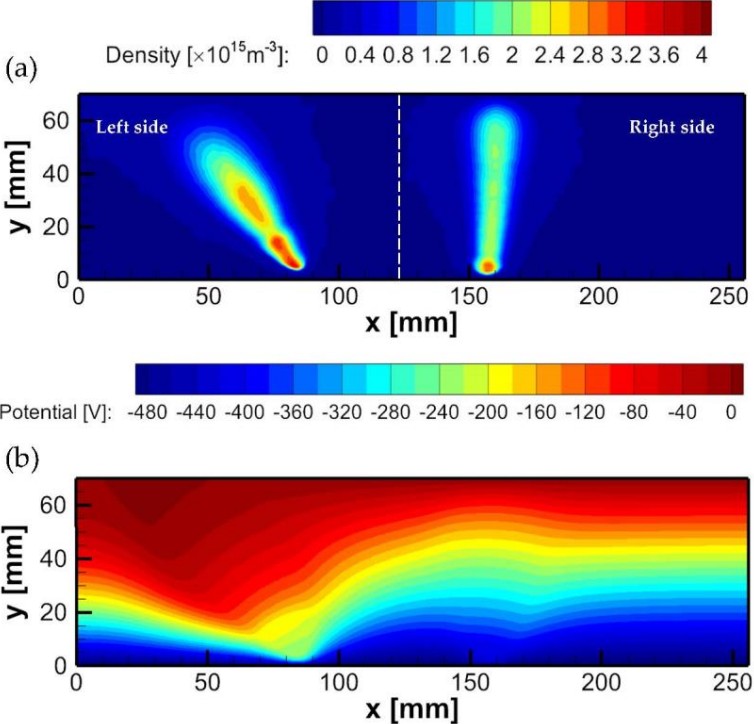

**Figure 9.** (**a**) Plasma density and (**b**) electric potential of the asymmetric magnetic flux density shown in Figure 8.

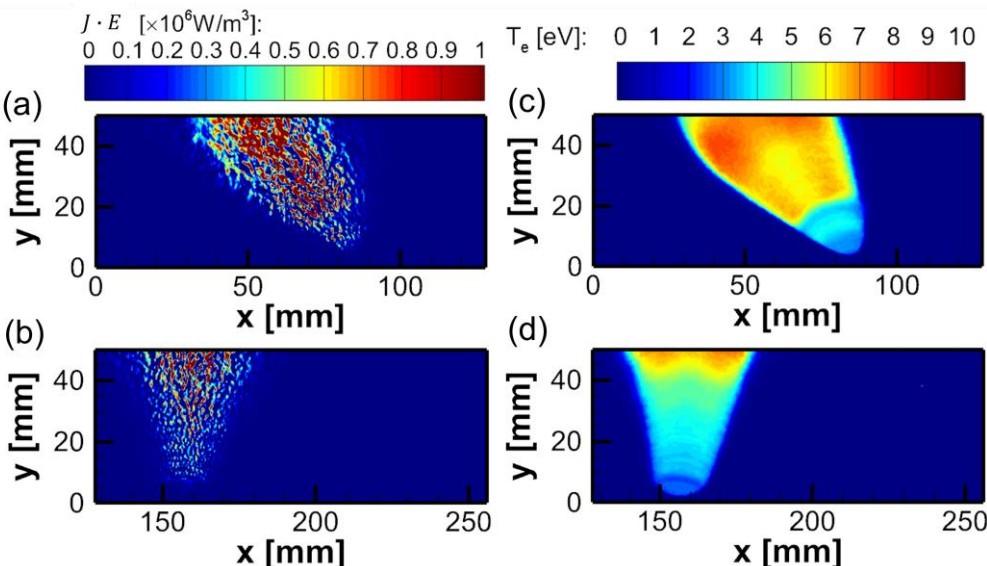

**Figure 10.** (**Left**) Profiles of $J \cdot E$ of the (**a**) left side and (**b**) right side of the domain for the asymmetric case. (**Right**) Profiles of electron temperatures of (**c**) the left side and (**d**) the right side of the domain in the asymmetric case.

The spatial distribution of plasma density on the right side seems more vertical to the target than on the left side. It is because magnetic field lines cause the difference in the spatial profiles of the plasma density. Plasmas are placed along the direction of $-\nabla B$, where $B$ is the magnitude of the magnetic flux density. The profiles of electron heating and electron temperatures shown in Figure 10 also show the same tendency toward less electron heating (Figure 10b) with more intensified magnetic fields. This result indicates that not only the magnitude of the magnetic flux density but also the magnetic field profile determine the transition of the striation regime. In addition, this kind of asymmetric configuration of magnets can be an option to control target erosion profiles since there will be different sputtering characteristics depending on the sputtering region on the target.

## 3. Three-Dimensional Simulation for Azimuthal Symmetry of Magnets

In Section 2, the 2D simulation was performed without consideration of the drift motion of electrons. As shown in Figure 11, the magnetic and electric fields normal to the target surface in the z-direction are crossed and generate an azimuthal E × B drift in the 3D space. In addition, $-\nabla B$ is also in the z-direction to cause a grad-B drift in the azimuthal direction. Finally, the density gradient in the z-direction also induces diamagnetic drift. These three types of drift motion play an essential role in transport in the azimuthal direction. Therefore, it is mandatory to include the effect of the drift motion under the variation of the curvature of the magnetic field lines.

The design of permanent magnets can vary depending on either the desired erosion region of the target plate or the characteristics of deposited thin films. For example, azimuthally asymmetric magnets make magnetic fields, and generated plasmas focus on a specific region to be sputtered, which can be used to make even more uniform erosion of the sputtering target [38,39]. This section discusses the effects of magnets' azimuthal symmetry on DCMS plasmas with three-dimensional electrostatic PIC-MCC simulation results. Detailed information, including the numerical method and the performance of the simulation code, was reported in the previous study [22].

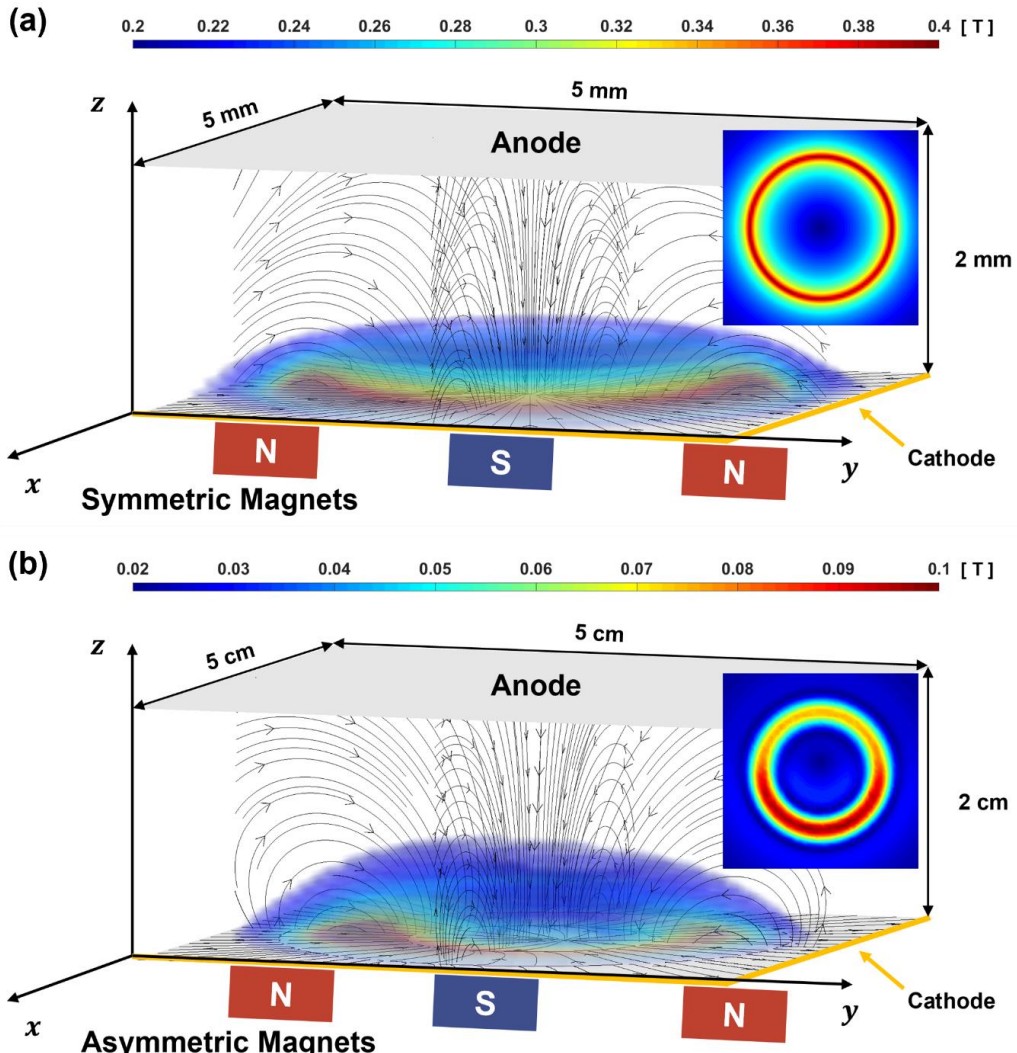

**Figure 11.** Schematic domains and profiles of magnetic fields in the three-dimensional simulation with (**a**) azimuthally symmetric magnets and (**b**) azimuthally asymmetric magnets.

Schematic diagrams of the 3D simulation domain with both azimuthally symmetric and asymmetric magnetic fields are depicted in Figure 11a,b, respectively. The simulation domains for both cases indicate simplified DCMS systems with boundaries that consist of one cathode surface and five grounded conductors. A fixed electric current is applied to the cathode in this 3D simulation instead of the constant electric power to obtain simulation results more quickly. Note that the simulation conditions differ since the results are obtained from different studies. For the symmetric case, the domain size is 5 mm × 5 mm × 2 mm, and the number of cells is 80 × 80 × 32. The applied current is 2 mA. The time step $\Delta t$ is $2 \times 10^{-11}$ s. The gas pressure is set to 150 mTorr. Argon gas is used and assumed to be uniform in the discharge region. For the asymmetric case, the domain size is 50 mm × 50 mm × 20 mm, and the number of cells is 64 × 64 × 32. The applied current is 20 mA. The time step $\Delta t$ is $1 \times 10^{-11}$ s. The gas pressure is set at 50 mTorr. It also considers uniform argon feed gas in the discharge region. The data were extracted at $r = 1.6$ mm and $r = 1.6$ cm, respectively, to investigate the change in the azimuth direction over time in the symmetric magnet and asymmetric magnet cases. These radii were set to follow the line of the maximum electron density when extracting data in the azimuth direction. The axial positions of the symmetric and asymmetric magnets were set to 0.75 mm and 3.125 mm, respectively, because the steady state results showed that the electron densities here had peak points on the two-dimensional xy plane.

Figure 12a presents the change in the simulated density of DCMS plasmas under the symmetric magnetic field shown in Figure 11a. Initially, the plasma density profile shows azimuthal symmetry at 0.75 ms. However, after 1 ms, it shows the evolution of the $m = 1$ mode. Finally, after 2.5 ms, the plasma density shows an $m = 2$ mode. Figure 12b shows the time evolution of the plasma density in the azimuthal region at the center, which clearly shows the growth of the m = 1 mode from 1 to 2 ms and the $m = 2$ mode after 2.5 ms. The symmetric magnets cause rotating structures of plasmas, which are well known as a phenomenon of rotating spoke instabilities [40–44]. In this case, the dominant mode is $m = 2$ in the steady state. The direction of rotation is clockwise, and the rotation velocity is 79.4 km/s. At the steady state, the $m = 2$ mode has an oscillation frequency of 3.03 MHz.

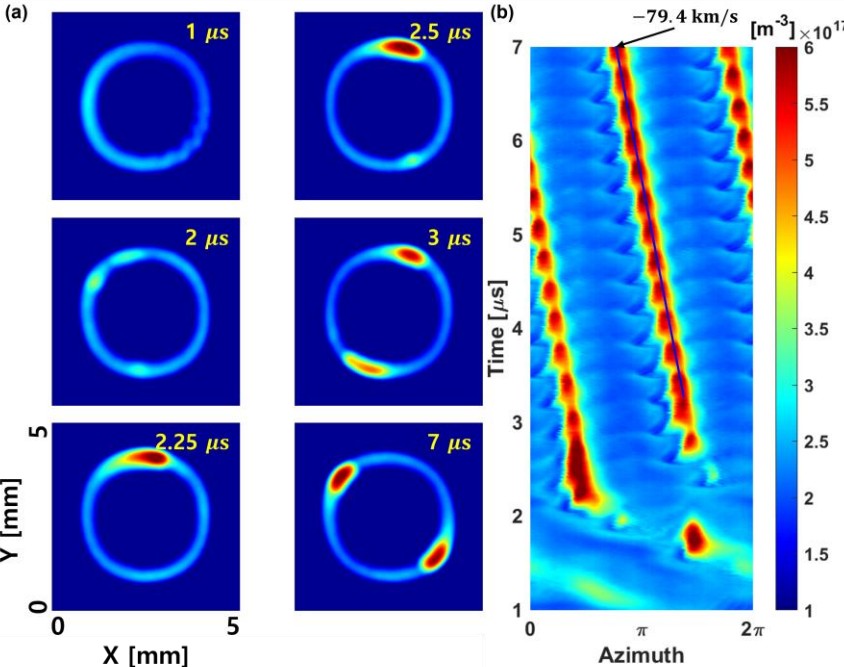

**Figure 12.** Time evolution of (**a**) spatial distributions of plasma density in xy plane with the symmetric magnet and (**b**) azimuthal distributions of the plasma density.

On the other hand, Figure 13 shows the plasma density profiles for the asymmetric magnetic fields shown in Figure 11b. The asymmetric magnetic field profile generates a quasi-stationary density structure with an azimuthal $m = 1$ mode. In the beginning, there is a transient time that shows a slight rotation of the density profile until $t < 4$ ms. However, the density pattern is almost static after 4 ms, following the change in the magnitude of the magnetic flux density. The high-density region is the same as where the azimuthal direction's drift velocity decreases.

Figure 14 shows the 3 types of azimuthal drift velocities measured at $z = 3.125$ mm and $r = 1.6$ cm for the steady state of Figure 13. The grad-B drift is the most dominant, while the E × B drift is relatively small and uniform. In addition, the direction of the grad-B drift is counter-clockwise, but that of the E × B drift is clockwise. Therefore, the most dominant factor triggering the instability is the gradient of the magnetic field profiles.

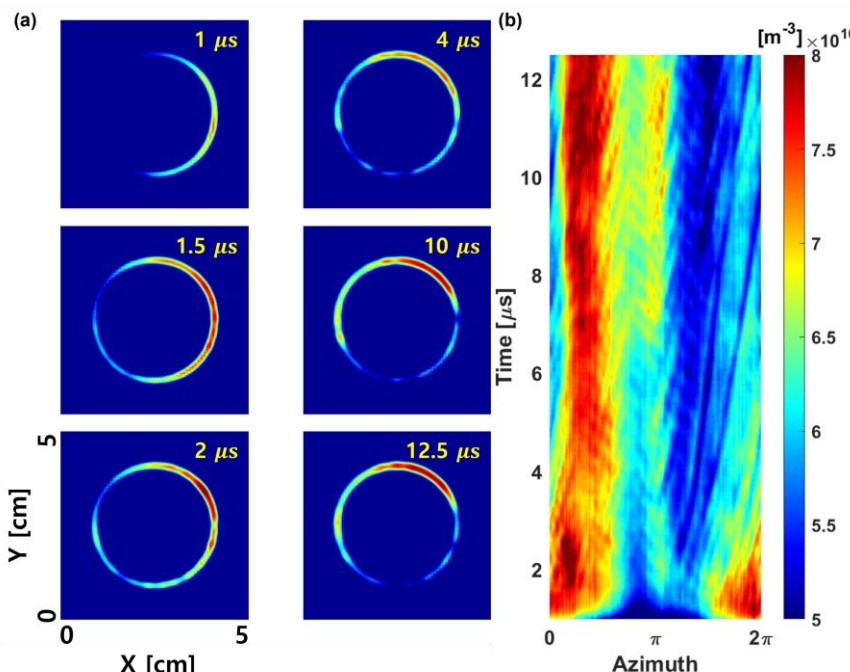

**Figure 13.** Time evolution of (**a**) spatial distributions of plasma density in xy plane with the asymmetric magnet and (**b**) azimuthal distributions of the plasma density.

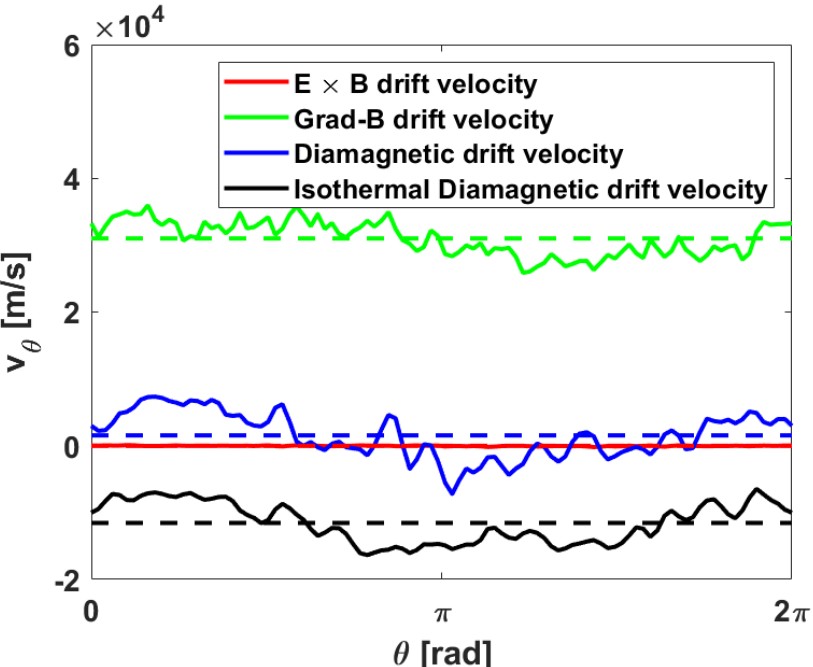

**Figure 14.** Azimuthal drift velocities were measured at $z$ = 3.125 mm and $r$ = 1.6 cm for E × B (red), grad-B (green), and diamagnetic (blue) drift, respectively. The black line is for the diamagnetic drift under the assumption of an isothermal plasma without a temperature gradient. Dashed lines indicate the mean values of each drift velocity.

The different states of each case can be analyzed with field energy. The electrostatic potential energy is given by $q\phi$ where $\phi$ is the electric potential. Figure 15 shows the time evolution of field energies for the two cases. The field energy of the symmetric case on a small scale oscillates since there are constant interactions between waves and charged particles, while that in the asymmetric case on a large scale saturates without oscillation. The reasons for the difference can be various since the simulation conditions

are also different. However, the fundamental mechanism of the stationary structure with asymmetric magnets is not fully understood yet.

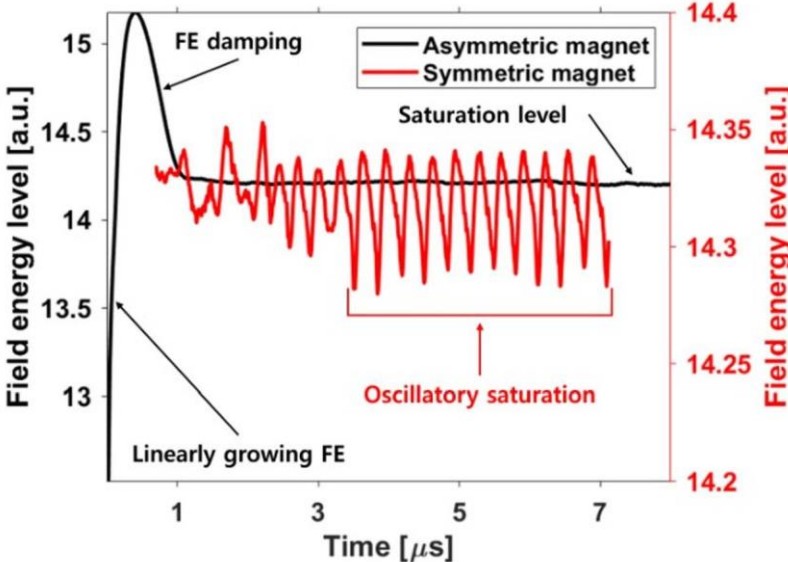

**Figure 15.** Saturation levels are shown in logarithmic scale for the normalized potential energy with an asymmetric (black) and a symmetric (red) magnet.

## 4. Conclusions

The effects of various configurations of permanent magnets on DCMS plasmas have been investigated with a 2D and a 3D PIC-MCC simulation. In the 2D PIC-MCC simulation, higher plasma densities with striations were obtained with a magnetic flux density of 350 G on the magnet surface than with 500 G. The increase in plasma density is caused by the enhanced electron heating under the weak magnetic flux density, although the electron confinement is much better under the strong magnetic flux density. The electron heating mechanism is related to wave-like transports along the magnetic field lines.

With the increase in the yoke thickness, the magnetic flux density increases slightly in the plasma region, and the geometry of the field lines changes. The plasma density near the target surface does not change much with the yoke thickness change. However, the density profile far from the target surface changes significantly. It indicates that more heated electrons propagate following the far magnetic field line but not near the target surface, where the strong magnetic field still limits the perpendicular transport of electrons. It shows the importance of the structure of magnetic field lines. Another example to show the importance of the magnetic field profile is the case with an asymmetric magnet structure. Even though the electron confinement is better where the gap from the north to the south pole is shorter, the plasma density is high in the weak-magnetic field region because the electron heating is enhanced.

Another interesting phenomenon of a stationary structure with azimuthally asymmetric magnets is investigated with a 3D PIC-MCC simulation. The difference between cases with azimuthally symmetric and asymmetric magnets is whether the azimuthal drift velocity is uniform. It was discussed along with the time evolution of the plasma density and the field energy. With the symmetric magnet on a small scale, oscillation patterns in the $m = 2$ mode are generated at an oscillation frequency of 3.03 MHz. However, with the asymmetric magnet on a larger scale, a static $m = 1$ mode lasts long without oscillation. We found that the grad-B drift is the most dominant from the measurement of the three drift velocities. Therefore, the instability generating the spoke shown in Figure 12 could be caused by the strong shear of the significant grad-B drift for a high curvature magnet. It is a topic for future work.

Although the number of specific simulation cases is limited in this paper, newly found fundamental influences of magnetic configurations on DCMS plasmas were investigated with kinetic approaches. However, the structures of magnets in a DCMS system can be much more complex depending on the desired characteristics of target sputtering or deposition. For example, additional permanent magnets or electromagnets can be applied to modify plasma distribution or improve the uniformity of the deposition profile. Therefore, more fundamental studies are necessary to determine the specific magnet design and configuration.

**Author Contributions:** Conceptualization, Y.H.J., C.C. and H.J.L.; Methodology, Y.H.J. and C.C.; Software, C.C. and H.P.; Validation, C.C. and H.J.L.; Formal analysis, Y.H.J., C.C. and H.P.; Investigation, Y.H.J., C.C. and H.P.; Data curation, Y.H.J., C.C. and H.P.; Writing—original draft, Y.H.J.; Writing—review and editing, H.J.L.; Visualization, C.C.; Supervision, H.J.L.; Project administration, H.J.L.; Funding acquisition, H.J.L. All authors have read and agreed to the published version of the manuscript.

**Funding:** This work was supported by the National Research Council of Science and Technology (NST) grant by the Korean government (MSIT) (No. CRC-20-01-NFRI) and by the National R&D Program through the National Research Foundation of Korea (NRF), funded by the Ministry of Education, Science and Technology (Grant No. NRF-2019R1A2C1088518).

**Institutional Review Board Statement:** Not applicable.

**Informed Consent Statement:** Not applicable.

**Data Availability Statement:** Not applicable.

**Conflicts of Interest:** The authors declare no conflict of interest.

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
