# Peer review of "Particle-in-Cell Simulations for the Improvement of the Target Erosion Uniformity by the Permanent Magnet Configuration of DC Magnetron Sputtering Systems"

_coatings, doi:10.3390/coatings13040749_

Round 1
Reviewer 1 Report
The objective of this manuscript is well described. It present PIC simulations devoted to improve target erosion uniformity in DC magnetron sputtering systems. The simulations was focused on the permanent magnet configuration (hence on the symmetry of the magnets, strength of field). 2D and 3D results have been presented. I found the number of figures a bit exagerated in the sense that I am not sure that all the figures are necessary. In my opinion the most critical point is that the results are presented and a conclusion is drawan in a complete disconnection from experiments. The manuscript does not mention how these results compare to experiment. I thnik this point is missing, at least the authors should mention how their results can be checked and if there are plans to do that.
Besides this point, there are some few typographical errors or some incorrect englih words or sentences. These are listed below:
-Page 2
*line 63. utilize-->used
*line 66. are bein utilized--> have been used
*line 78. mentioned--> presented
-Pgae 3.
In figure 2, the color scales are different in a) and b) The maxima are different. This deserves to be indicated.
*line 107. shown-->seen
*line 119. Figure 2a-->Figure 2.
-Page 4
*line 174. It is not indicated how this velocity value is obtained.
-Page 6.
* numbers in the figure represent dimensions (lengths), this should be indicated as well as the unit (mm).
-Page 8.
* In figure 4, \vec{j} \dot \vec{E} is used with a unit representing a power density. It is not indicated what does stand for. Also the unit of the temperature is not indicated.
* Figure 5. Same remark for the numbers and the units as for figures 1 and 2. Same for other figures.
Eventually I think that the figures are not sufficiently interpreted.
Author Response
Thank you for the helpful comments by the referee.
We added more content (more than 40% of the whole manuscript) to describe the details of the explanation for the figures. As the realistic DCMS chamber is much larger than the simulated case, we could not compare our simulation results with a realistic chamber. However, the observed results agree well with the reported experimental results qualitatively.
We revised the manuscript following the comments as well.
Reviewer 2 Report
This manuscript has studied the permanent magnets on DCMS plasmas by using the PIC-MCC simulations. The authors found that higher plasma densities with striations can be obtained with weaker magnetic fields according to the two-dimensional PIC-MCC simulation results. This is important study for fabrication film using the magnetron sputtering device. However, the major revision is necessary before publication.
1. The authors should introduce the importance of the plasma in the Introduction section.
2. All the figures should be discussed in detail, the authors should add more discussions about figures.
Author Response
Thank you for the helpful comments by the referee.
Following comment #1, we revised the introduction to address the importance of the plasma effect by the change of the magnetic field profiles.
Following comment #2, all the figures are discussed in detail, and more than 40% of the total contents are newly added in the revised manuscript.
Reviewer 3 Report
1. First of all the paper must be design according to the journals guidelines. Please revise it.
2. All figures should be placed in the text body.
3. The novelty of the presented research should be highlighted in the Introduction part.
4. The information about practical application of the research should be inserted in the text.
5. References should be formatted as:
Author 1, A.B.; Author 2, C.D. Title of the article. Abbreviated Journal Name Year, Volume, page range.
6. The Conclusion part is too short, please improve it.
7. There are some insufficient typos and English mistakes in the text.
8. The Introduction part should be improved by relevant literature in the field of magnetron deposition: https://doi.org/10.3390/ma15217770, https://doi.org/10.3390/coatings12121807
But any way I impressed by this paper. But authors must explain some details and improve the paper in accordance with my comments. The paper should be sent to me for the second analysis after the minor revisions.
Author Response
Thank you for the helpful comments by the referee. Please see the answers as follows.
- We revised the manuscript under the journal guideline. If something is still missing in the present version, the publisher team will help us how to change the format.
- The figures will be placed in the text body when it is published finally.
- The introduction was fully revised to emphasize the novelty of the presented work.
- More than 40% of the manuscript was revised to include concrete physical interpretation.
- The reference format will be edited following the publisher's guide, after all.
- The conclusion part has been improved. Please see the highlighted sentences in red.
- The introduction also has been improved with new references added as [4, 5] and [38].
Reviewer 4 Report
Magnetron sputtering is a very established technique. Most of the literature is very old. The time scale chosen, like the difference in time is 0.25 microseconds which seem too unreasonable to have such control and estimation. The novelty is not clear and does not seem to be attracting much attention to researchers in the field.
Author Response
Thank you for the helpful comments by the referee.
Even though magnetron sputtering is a well-established technology, it is still not well known how the magnetic field structure affects the plasma density and how the instability is triggered. In this study, we investigated the effect of the magnetic field structure and the drift motion of the electrons using 2D and 3D PIC simulations. New physics from the PIC simulation results are explained in the revised manuscript (more than 40% of the whole manuscript was revised).
The simulation results were obtained at a steady state of the PIC simulations. That means that the same phenomena keep going in the system.
Please read the revised version and evaluate the advantage of the state-of-the-art 3D simulation results.
Round 2
Reviewer 1 Report
The manuscript is far better than the original version. It has been improved and deserve now to be accepted for publication in this journal. There are few minor typographical errors to correct. Some of them are below:
- in the abstract, ExB is a vectorial product so E and B should appear as vectors using bold for instance.
-line 24. In the sentence "The atoms transport through the chamber before...", transport is used as a verb. You may use the verb move for instance.
-line 59: magnetized electron --> magnetized electrons.
Author Response
We appreciate the referee's comments once again. We revised the manuscript following the suggestion.
Reviewer 2 Report
The author have revised the manuscript according to the Reviewer's comments. I recommend it can be published.
Author Response
Thank you very much for the final decision of acceptance.
Reviewer 4 Report
Has been improved greatly, may be accepted
Author Response

(The authors gave the same response as above.)
